# Socioeconomic and geographical inequalities in adolescent fertility rates in Sierra Leone, 2008–2019

**Augustus Osborne**[1]*, **Camilla Bangura**[1], **Bright Opoku Ahinkorah**[2,3]

**1** Department of Biological Sciences, School of Basic Sciences, Njala University, PMB, Freetown, Sierra Leone, **2** REMS Consultancy Services Limited, Sekondi-Takoradi, Western Region, Ghana, **3** Faculty of Health and Medical Sciences, The University of Adelaide, Adelaide, Australia

\* augustusosborne2@gmail.com

## Abstract

### Background

Sierra Leone, like many other sub-Saharan African countries, grapples with the challenge of high adolescent fertility rates. This study examines the socio-economic and geographical inequalities in adolescent fertility rates in Sierra Leone between 2008 and 2019.

### Methods

Three rounds of the Sierra Leone Demographic and Health Surveys (2008, 2013, and 2019) were analysed to examine inequalities in adolescent fertility rates. Descriptive analyses were performed using the online version of the World Health Organization's Health Equity Assessment Toolkit software. Adolescent fertility rate was stratified using four dimensions: economic status, education, place of residence, and province. Difference (D), ratio (R), population attributable risk (PAR) and population attributable fraction (PAF) were calculated as measures of inequality.

### Results

The adolescent fertility rates in Sierra Leone declined from 142.5 births per 1,000 women aged 15–19 years in 2008 to 103.5 births per 1,000 women aged 15–19 years in 2019. For economic status, inequality in adolescent fertility rates decreased from 117.3 births per 1,000 adolescent girls in 2008 to 110.6 in 2019. The PAF indicated that the national adolescent fertility rate could have been 46.8% lower in 2008, 42.5% lower in 2013, and 53.5% lower in 2019 if all wealth quintiles had the same fertility rates as the wealthiest quintile (quintile 5). Educational inequality in adolescent fertility rates decreased significantly, from 135.3 births per 1,000 adolescent girls in 2008 to 75.8 in 2019. The PAF showed that the setting average of adolescent fertility rate could have been 57.9% lower in 2008, 33.1% lower in 2013, and 23.9% lower in 2019 without education-related disparities. For place of residence, inequality between urban and rural areas decreased from 82.3 births per 1,000 adolescent girls in 2008 to 74.5 in 2019. The PAF indicated that the national adolescent fertility rate

**Data Availability Statement:** The dataset utilized in this study can be accessed through the WHO Equity website at https://whoequity.shinyapps.io/heat/. Interested researchers can obtain the minimal datasets by visiting this link, where they

will find relevant titles and descriptions of the data available. To ensure immediate access, users should follow the instructions provided on the website. Additionally, the methods outlined in our study can be replicated using the datasets available from this source, allowing for the verification of our findings. We confirm that the authors did not have any special access privileges to the data beyond what is available to the public, ensuring that all researchers have equal opportunity to access and utilize the datasets for their investigations.

**Funding:** The author(s) received no specific funding for this work.

**Competing interests:** The authors have declared that no competing interests exist.

**Abbreviations:** D, Difference; HEAT, Health Equity Assessment Toolkit; PAF, Population Attributable Fraction; PAR, Population Attributable Risk; R, Ratio; SDG, Sustainable Development Goal; SLDHS, Sierra Leone Demographic and Health Survey; STROBE, Strengthening the Reporting of Observational Studies in Epidemiology; WHO, World Health Organization.

could have been 32.9% lower in 2008, 30.7% lower in 2013, and 33.9% lower in 2019 if rural girls had the same fertility rates as urban girls. Our results further showed that inequality based on province decreased from 77.9 births per 1,000 adolescent girls in 2008 to 64.0 in 2019. The PAF showed that the national average of adolescent fertility rates could have been 34.6% lower in 2008, 37.6% lower in 2013, and 35.8% lower in 2019 without provincial disparities.

## Conclusion

Our study found a positive decline in AFR across socioeconomic and geographic groups in Sierra Leone, but significant inequalities remain. Economic status and education are key drivers, with the poorest quintile consistently showing higher AFR. Although AFR declined among girls across all levels of education over time, it increased between 2008 and 2019 for those with primary and higher education. Rural areas had a higher AFR than urban ones, though with less national impact. Policymakers should focus on improving economic opportunities, enhancing quality education, and expanding access to family planning services to reduce adolescent pregnancy and address socioeconomic and educational inequalities.

## Introduction

Adolescent fertility has decreased worldwide in the last twenty years. However, reductions in several low-and middle-income countries (LMICs) have been minimal [1, 2]. Adolescent fertility rate (AFR) measures the number of births that occur each year to girls between the ages of 15 and 19 years, expressed as the number of births per 1,000 women [3, 4]. By 2030, there is a projected increase in adolescent pregnancy worldwide, which can be attributed to the anticipated expansion in the overall adolescent population [5]. Adolescent pregnancy rates remain elevated in numerous countries [6], resulting in approximately 21 million adolescent girls becoming pregnant each year in LMICs [7]. Out of this number of pregnancies, 50% are unintended and lead to around 12 million births [7–9]. Furthermore, 55% of unwanted pregnancies among adolescents result in unsafe abortions. These unsafe abortions are widespread in LMICs due to the limited availability of information and services on adolescent sexual and reproductive health [7, 8].

Adolescent pregnancies have been extensively studied and have been found to have negative consequences for both adolescent mothers [7, 10] and their kids [7, 11–13]. The adverse effects on the mother and child become more severe when the adolescent mother's age decreases [14, 15]. Moreover, pregnancy-related complications are significant contributors to mortality among adolescent girls worldwide [15, 16]. Approximately 4 million adolescents are exposed to the dangers of unsafe abortions, which frequently result in maternal illness and death [9]. Furthermore, adolescent pregnancies have a detrimental impact on other adolescent students, acquaintances, relatives, and the community to which the pregnant adolescent belongs [17].

Sierra Leone has experienced a consistent decrease in the total fertility rate (TFR) throughout the years, dropping from 5.1 children per woman in 2008 to 4.9 in 2013 and further to 4.2 in 2019 [18]. Women in both rural and urban regions have experienced a fall in numbers. In rural areas, the decline has been from 5.8 in 2008 to 5.7 in 2013, and 5.1 in 2019, while in urban areas, it has been from 3.8 in 2008 to 3.5 in 2013, and 3.1 in 2019. The fertility rate is

lowest among women aged 15–19 (102 births per 1,000 women), and highest among those aged 20–24 (196 births per 1,000 women) [18].

Despite the relatively lower fertility rates among adolescent girls in Sierra Leone, the country is working to address adolescent fertility due to the negative effects it has on teen mothers and their babies [19]. Interventions to increase access to family planning services and adolescent education have been implemented to empower adolescent girls to make informed reproductive health choices [20]. The National Adolescent Health Policy (2016) outlines a framework for improving adolescent health, including sexual and reproductive health [21]. In addition, the Free Quality Education Policy aims to remove financial barriers to girls' education [20]. Despite these improvements, many girls, particularly those in rural areas, have limited access to education and family planning services [20]. Early marriage and childbearing are ingrained in some cultures, and social pressure to get pregnant and give birth can be intense [21]. Poverty also limits girls' educational opportunities and increases their vulnerability to transactional sex or pressure to marry young to ease the financial burden on families [19]. Continued efforts are needed to improve access to education family planning and address cultural norms.

Various socioeconomic, cultural, environmental, and health service-related factors such as sexual coercion, limited adoption and utilisation of contraceptives, insufficient parental communication and support, early marriage, negative religious beliefs, early initiation of sexual activity, living in rural areas, unfavourable cultural attitudes, diminished self-confidence, and low educational attainment have been found to significantly predict adolescent pregnancies and deliveries in sub-Saharan Africa (SSA) [22–24]. Moreover, insufficient sexuality education, societal expectations to marry and give birth, and limited availability of contraception and reproductive healthcare facilities all contribute to the occurrence of teenage pregnancies [7, 25, 26]. Prior research conducted in Ghana [27], Ethiopia [28], and several regions of Africa [24] have documented disparities in AFR, with impoverished teenage girls, individuals without education, and those living in rural locations exhibiting greater fertility rates. Therefore, the educational level of mothers, their socioeconomic situation, and the area where they live play a crucial role in determining the rates of adolescent pregnancies and their subsequent outcomes [29].

Like many sub-Saharan African countries, Sierra Leone grapples with the challenge of AFR. While the nation has made strides in recent decades, significant disparities persist across socioeconomic and geographical lines. The country emerged from a brutal civil war in 2002 and embarked on a reconstruction and economic growth period. However, the scars of conflict left lasting social and economic inequalities that could potentially affect AFR. Understanding these inequalities is crucial for designing effective policies to reduce AFR and improve reproductive health outcomes for young people. This study examines the socio-economic and geographical inequalities in AFR in Sierra Leone between 2008 and 2019. By examining these inequalities, we can identify areas for targeted interventions to reduce AFR and improve reproductive health outcomes for all young people in Sierra Leone.

## Methods

### Study setting and data source

We used data from the 2008, 2013, and 2019 Sierra Leone Demographic and Health Surveys (SLDHS). The SLDHS is a nationwide survey that aims to identify consistent trends and changes in demographic indicators, health indicators, and social issues among individuals of all genders and age groups. A detailed description of the study design and sampling methodology can be found in the SLDHS report [18]. This study included adolescents aged 15–19 in the corresponding SLDHS cycles. The 2008, 2013, and 2019 SLDHS data were available for direct

use on the online version of the WHO's Health Equity Assessment (HEAT) Toolkit [30]. We wrote this paper in accordance with Strengthening the Reporting of Observational Studies in Epidemiology (STROBE) guidelines [31].

## Inequality measures

The variable for which inequality was assessed in this study was AFR. This variable is calculated as the rate of births per 1,000 females between the ages of 15 and 19. We used four dimensions to examine the extent of inequality. These were economic status, educational level, place of residence, and subnational province based on previous studies [27, 28, 32]. We assessed economic status using wealth index. Within the DHS, wealth index is calculated by analysing various family ownerships and attributes using the Principal Component Analysis (PCA) technique [33]. It is divided into five categories called quintiles: Quintile 1 (poorest), 2 (poorer), 3 (middle), 4 (richer), and quintile 5 (richest). The educational level of the women was categorised into three groups: no education, primary, secondary/higher. Place of residence was categorized into urban or rural, and subnational province as Eastern, Northern, Northwestern, Southern, and Western.

## Statistical analysis

The analysis was performed on the online version of the HEAT toolkit developed by the WHO [30], which examines health disparities within and between nations across various health indicators and socioeconomic issues, including child, maternal, and reproductive health [34]. This study focused on four inequality measures: difference (D), ratio (R), population-attributable fraction (PAF), and population-attributable risk (PAR). In this study, D measures the absolute gap in AFR between two subgroups, providing a direct comparison of their respective rates. For a binary dimension such as place of residence, D is calculated as AFR in rural areas—AFR in urban areas. For non-ordered dimensions such as level of education, D is calculated as the category of the educational level with the highest AFR—the category with the lowest AFR. If there is no inequality, D takes the value zero. Greater absolute values indicate higher levels of inequality. R compares the AFR of two subgroups by dividing the AFR of one subgroup by the other, offering a relative measure of inequality. For a binary dimension such as place of residence, R is calculated as AFR in rural areas/AFR in urban areas. For non-ordered dimensions such as level of education, R is calculated as the category of the educational level with the highest AFR/the category with the lowest AFR. If there is no inequality, R takes the value one. R takes only positive values. The further the value of R from one, the higher the level of inequality. Both D and R are unweighted measures (simple measures), meaning they do not account for the population sizes of the subgroups and focus solely on the two groups being compared. On the other hand, both PAF and PAR reflect the potential improvement in the average level of AFR that could be achieved if all population subgroups matched the level of AFR seen in a reference group. Considering that AFR is an adverse indicator, PAF and PAR yield negative values. PAF and PAR equals zero when no further improvement is possible, meaning that all subgroups of a dimension have attained the same level of the AFR as the reference subgroup. PAF and PAR are complex measures since they are weighted and account for the population sizes. In addition, both D and PAR are absolute measures of inequality and R and PAF are relative measures. The literature highlights the calculation and analysis of these measures [34, 35].

## Ethics approval statement

This study did not seek ethical clearance since the online version of the WHO HEAT toolkit is freely available in the public domain.

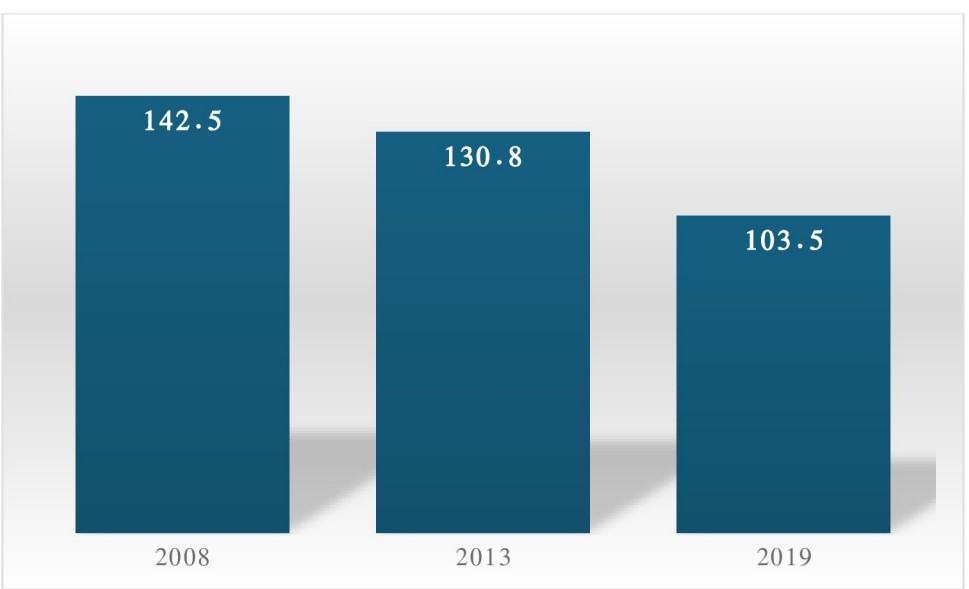

**Fig 1. Trends in adolescent fertility rates in Sierra Leone, 2008–2019.**

## Results

Fig 1 shows the trends in AFR across the three survey years in Sierra Leone. The AFR declined between 2008 (142.5 births per 1,000 women aged 15–19 years) and 2019 (103.5 births per 1,000 women aged 15–19 years).

### Trends in adolescent fertility rates in Sierra Leone by different inequality dimensions, 2008–2019

Table 1 shows the AFR in Sierra Leone from 2008 to 2019. Over the years, AFR decreased across all categories of economic status, with the most significant drop seen among adolescent girls in quintile 4, from 137.5 births per 1,000 women aged 15–19 years in 2008 to 83.8 births in 2019. In terms of education level, whereas AFR decreased from 195.2 birth per 1,000 women aged 15–19 years in 2008 to 154.6 births in 2019 among adolescent girls with no formal education, AFR increased from 151.8 births per 1,000 women aged 15–19 years in 2008 to 152.9 births in 2019 among those with primary education. In addition, there was a significant increase in AFR from 59.9 births per 1,000 women aged 15–19 years in 2008 to 78.7 births in 2019. AFR reduced between 2008 and 2019 in both rural and urban areas, with a much higher decrease in rural areas (177.9 births per 1,000 women aged 15–19 years in 2008 to 142.9 in 2019). In all five provinces, AFR reduced over time, with the Northern province experiencing the highest rate of reduction.

### Inequality measures of adolescent fertility rates in Sierra Leone, 2008–2019

Table 2 shows the inequality in factors associated with AFR, using various indices from 2008 to 2019 in Sierra Leone. For brevity of presentation, we report results for one simple measure (D) and one complex measure (PAF). Both measures also represent both absolute (D) and relative measures (PAF). For economic status, the results from the inequality measure D showed that there was a decrease from 117.3 births per 1,000 women aged 15–19 years in 2008 to 110.6

**Table 1. Trends in adolescent fertility rates in Sierra Leone by different inequality dimensions, 2008–2019.**

| Dimension | Sample | Births per p1,000 women aged 15–19 years (95% CI) | Sample | Births per p1,000 women aged 15–19 years (95% CI) | Sample | Births per p1,000 women aged 15–19 years (95% CI) |
|---|---|---|---|---|---|---|
| | | 2008 | | 2013 | | 2019 |
| **Economic status** | | | | | | |
| Quintile 1 (poorest) | 1086 | 193.1 (140.3, 259.8) | 2737 | 169.8 (151.7, 189.5) | 1937 | 158.7 (143.7, 175.0) |
| Quintile 2 | 1045 | 176.6 (136.1, 226.0) | 2577 | 160.8 (127.1, 201.5) | 2455 | 148.1 (134.5, 162.8) |
| Quintile 3 | 1140 | 177.8 (140.8, 222.0) | 2933 | 157.3 (131.4, 187.3) | 2881 | 130.9 (118.2, 144.7) |
| Quintile 4 | 1369 | 137.5 (110.6, 169.8) | 3650 | 132.3 (113.4, 153.7) | 3931 | 83.8 (75.5, 93.0) |
| Quintile 5 (richest) | 1861 | 75.8 (60.8, 94.1) | 4804 | 75.1 (60.3, 93.1) | 3928 | 48.1 (40.0, 57.7) |
| **Education** | | | | | | |
| No education | 3189 | 195.2 (159.6, 236.5) | 4958 | 190.0 (176.3, 204.5) | 3002 | 154.6 (141.5, 168.6) |
| Primary education | 1149 | 151.8 (114.7, 198.2) | 2657 | 168.3 (134.8, 208.1) | 1994 | 152.9 (137.7, 169.5) |
| Secondary or higher education | 2162 | 59.9 (44.9, 79.4) | 9087 | 87.5 (73.6, 103.7) | 10137 | 78.7 (72.2, 85.7) |
| **Place of residence** | | | | | | |
| Rural | 3708 | 177.9 (164.0, 192.7) | 9716 | 159.7 (147.5, 172.7) | 7143 | 142.9 (133.6, 152.7) |
| Urban | 2794 | 95.5 (84.2, 108.2) | 6986 | 90.5 (81.2, 100.9) | 7990 | 68.3 (62.1, 75.2) |
| **Province** | | | | | | |
| Eastern | 1136 | 153.8 (134.0, 175.8) | 3155 | 161.4 (139.8, 185.5) | 2827 | 115.9 (100.2, 133.7) |
| Northern | 2585 | 152.0 (134.4, 171.5) | 63989 | 131.7 (117.0, 147.9) | 3006 | 93.4 (82.1, 105.9) |
| Northwestern | No data | No data | No data | No data | 2587 | 130.4 (114.7, 147.9) |
| Southern | 1284 | 170.9 (143.4, 202.6) | 3346 | 156.2 (141.6, 171.9) | 2750 | 130.0 (113.5, 148.6) |
| Western | 1496 | 93.0 (75.6, 114.0) | 3802 | 81.5 (68.5, 96.8) | 3960 | 66.4 (57.3, 76.8) |

births per 1,000 women aged 15–19 years in 2019. The PAF revealed that the national average of adolescent fertility rate could have been 46.8 percent lower in 2008, 42.5 percent lower in 2013 and 53.5 percent lower in 2019 if all quintiles had the same fertility rates as quintile 5. Educational inequality decreased from a difference of 135.3 births per 1,000 women aged 15–19 years in 2008 to 75.8 births per 1,000 women aged 15–19 years in 2019. As shown from the PAF, that the setting average of adolescent fertility rate could have been 57.9 percent lower in 2008, 33.1 percent lower in 2013 and 23.9 percent lower in 2019 without education-related inequality. For place of residence, there was a decrease in inequality from a difference of 82.3 births per 1,000 women aged 15–19 years in 2008 to 74.5 births per 1,000 women aged 15–19 years in 2019. The PAF showed that the national average of adolescent fertility rate could have been 32.9 percent lower in 2008, 30.7 percent lower in 2013 and 33.9 percent lower in 2019 if adolescent girls in rural areas had the same fertility rates as those in urban areas. Province-based inequality decreased from a difference of 77.9 births per 1,000 women aged 15–19 years in 2008 to 64.0 births per 1,000 women aged 15–19 years in 2019. The PAF revealed that the setting average of adolescent fertility rate could have been 34.6 percent lower in 2008, 37.6 percent lower in 2013 and 35.8 percent lower in 2019 if there was no provincial inequality.

## Discussion

The study examines the socio-economic and geographical inequalities in AFR in Sierra Leone between 2008 and 2019. It was found that AFR has declined in Sierra Leone between 2008 (142.5 births per 1,000 women aged 15–19 years) and 2019 (103.5 births per 1,000 women aged 15–19 years). The study found that economic status and education are key drivers of

**Table 2. Inequality measures of adolescent fertility rates in Sierra Leone, 2008–2019.**

| | 2008 | | | 2013 | | | 2019 | | |
|---|---|---|---|---|---|---|---|---|---|
| Dimension | Estimate | Lower Bound | Upper Bound | Estimate | Lower Bound | Upper Bound | Estimate | Lower Bound | Upper Bound |
| **Economic status** | | | | | | | | | |
| D | 117.3 | 55.4 | 179.1 | 94.6 | 69.7 | 119.5 | 110.6 | 92.6 | 128.5 |
| PAF | -46.8 | -46.8 | -46.7 | -42.5 | -42.6 | -42.5 | -53.5 | -53.5 | -53.4 |
| PAR | -66.7 | -77.7 | -55.7 | -55.6 | -62.4 | -48.9 | -55.4 | -61.6 | -49.1 |
| R | 2.5 | 1.7 | 3.7 | 2.2 | 1.7 | 2.8 | 3.2 | 2.6 | 4.0 |
| **Education** | | | | | | | | | |
| D | 135.3 | 93.3 | 177.3 | 102.5 | 81.9 | 123.1 | 75.8 | 60.7 | 91.0 |
| PAF | -57.9 | -58.0 | -57.9 | -33.1 | -33.1 | -33.0 | -23.9 | -24.0 | -23.9 |
| PAR | -82.6 | -91.8 | -73.4 | -43.3 | -47.8 | -38.7 | -24.8 | -28.4 | -21.2 |
| R | 3.2 | 2.3 | 4.6 | 2.1 | 1.8 | 2.6 | 1.9 | 1.7 | 2.2 |
| **Place of residence** | | | | | | | | | |
| D | 82.3 | 63.6 | 101.0 | 69.1 | 53.2 | 85.1 | 74.5 | 62.9 | 86.0 |
| PAF | -32.9 | -33.0 | -32.8 | -30.7 | -30.8 | -30.7 | -33.9 | -34.0 | -33.9 |
| PAR | -46.9 | -56.0 | -37.8 | -40.2 | -45.8 | -34.6 | -35.1 | -39.5 | -30.7 |
| R | 1.8 | 1.6 | 2.1 | 1.7 | 1.5 | 2.0 | 2.0 | 1.8 | 2.3 |
| **Province** | | | | | | | | | |
| D | 77.9 | 42.7 | 113.0 | 79.8 | 53.0 | 106.6 | 64.0 | 44.8 | 83.2 |
| PAF | -34.6 | -34.7 | -34.5 | -37.6 | -37.7 | -37.5 | -35.8 | -35.9 | -35.7 |
| PAR | -49.4 | -62.9 | -35.9 | -49.2 | -57.2 | -41.2 | -37.1 | -44.1 | -30.0 |
| R | 1.8 | 1.4 | 2.4 | 1.9 | 1.5 | 2.4 | 1.9 | 1.6 | 2.3 |

D = Difference; PAF = Population Attributable Fraction; PAR = Population Attributable Risk; R = Ratio

inequalities in AFR. Rural areas had a higher AFR than urban areas, and there were provincial variations in AFR. However, the impact of these dimensions on overall inequality was less significant.

Findings on the decline in AFR in Sierra Leone may be due to increased access to education, especially for girls, as education can delay marriage and childbearing and empower young women to make informed choices about their reproductive health [21]. Improved family planning services, such as greater availability of contraceptives and family planning advice, can allow young women to plan their pregnancies [36]. Cultural attitudes towards teenage pregnancy may be changing, with increased awareness of its risks for both mother and child [24].

The findings of our study showed a decrease in inequality in AFR across economic status, with a D of 117.3 births per 1,000 women aged 15–19 years in 2008, which declined to 110.6 births per 1,000 women aged 15–19 years in 2019. However, the decline was minimal among adolescents in poorer households compared to wealthier households. These findings align with those of previous studies [37–39]. Poorer families may be less likely to prioritise education, especially for girls, limiting their knowledge about reproductive health and family planning options [40, 41]. Adolescents from poorer households may have less access to information campaigns and educational programs about sex education, contraception, and the risks of early pregnancy [42]. In poorer households, girls may be pressured to leave school early and enter the workforce, increasing their vulnerability to unplanned pregnancies. Poverty can be a driver of child marriage, where girls are married off young and expected to start families sooner [21]. To address this disparity, governments and policymakers should increase

access to education by promoting girls' education. Expanding access to affordable contraceptives and family planning services in low-income communities is crucial. Educational campaigns within communities can address cultural norms and promote healthy reproductive choices. By tackling these factors, Sierra Leone can work towards closing the gap in adolescent fertility rates between different socioeconomic groups.

Our findings also showed that whereas inequalities in level of education in general in general reduced over time, AFR increased among adolescent girls with both primary and secondary/higher levels of education between 2008 and 2019. The findings of a decrease in inequalities over time by level of education aligns with the findings of previous studies [27, 32, 37, 43]. With less education, young people might have limited knowledge about reproductive health and family planning options [44]. This can lead to unplanned pregnancies, contributing to higher AFR. Lower education often translates to lower socioeconomic status [21]. This can lead to factors like child marriage. In Sierra Leone, child marriage is still a concern. Less educated families might marry young daughters to ease financial burdens [21]. Traditional ideas about gender roles and family size might be more prevalent in areas with lower education levels [45]. These norms can influence AFR. Addressing the issue involves programs that educate young people, particularly girls, about sexual and reproductive health, which can significantly reduce AFR. By prioritising education, especially for girls, Sierra Leone can empower young women and improve their socioeconomic status, decreasing AFR. However, such education should be provided with caution considering the increase in adolescent rates between 2008 and 2019 among adolescent girls with at least primary education.

Furthermore, adolescents in rural Sierra Leone had a higher AFR than their urban counterparts. This finding is in line with the previous studies [27, 32, 37]. Rural areas often have fewer schools and educational resources. This can limit access to comprehensive sexual education, including information about family planning methods [36]. Rural communities in Sierra Leone often face greater poverty and economic hardship. Traditional practices that emphasise early marriage and childbearing might be stronger in rural areas [46]. Social pressure to conform to these norms can be high. Rural areas often have fewer healthcare facilities and qualified medical professionals. This can make it difficult for adolescents to access reproductive healthcare services, including contraception [47]. On the other hand, urban environments tend to have better access to education, healthcare, and economic opportunities [48]. This can empower young women and give them the knowledge and resources to delay childbearing. To address this disparity, the government and policymakers should expand access to quality education, particularly for girls, in rural areas, which is crucial. Investing in rural healthcare infrastructure and training healthcare professionals in sexual and reproductive health is essential. Community outreach programs can raise awareness about family planning and the benefits of delaying childbearing. By tackling these factors, Sierra Leone can work towards closing the AFR gap between rural and urban areas. This can also be expanded across the various provinces, especially in places where the reduction in AFR over time has been minimal.

## Policy and practice implications

The findings of our study on highlight the need for multi-dimensional policy and practice interventions to address the persistent socioeconomic and geographic disparities. The government and policymakers should partner with non-governmental organisations and community organisations to develop economic opportunities for young women through skills training in agriculture, entrepreneurship, or handicrafts. They should increase government funding for primary and secondary education, particularly for girls, and address barriers to school attendance, such as school fees or lack of childcare. These stakeholders should also develop and

implement comprehensive sexuality education programs within the school curriculum, ensuring age-appropriate information and addressing cultural sensitivities. There should also be training for teachers on delivering these programs effectively. In addition, government and policymakers should increase budgetary allocation to expand access to affordable, high-quality family planning services and contraceptives nationwide and train healthcare providers, including midwives and community health workers, to counsel adolescents about family planning options and provide confidential services—partner with local community leaders to address stigma and misinformation surrounding contraception. Also, partnership between local NGOs and community leaders should be established to develop culturally appropriate awareness campaigns about the risks of early pregnancy and the benefits of family planning and utilise community radio and local communication channels for outreach. By implementing these policy and practice changes, Sierra Leone can address the socioeconomic and geographic disparities in AFR and achieve a more equitable decline across the nation.

## Strengths and limitations

The DHS uses standardised methods across countries, allowing for comparisons within Sierra Leone and potentially with other countries over time. The surveys collect detailed information on socio-economic factors like household wealth, education, and place of residence, which can be linked to fertility rates. The WHO HEAT software is designed to help users explore and visualise inequalities in health data, making it a good fit for the topic. HEAT can calculate various inequality metrics, quantifying the disparities in adolescent fertility rates. In terms of limitations, the DHS uses a cross-sectional design, making it difficult to establish causal relationships between factors and AFR. Sensitive topics like adolescent fertility might be underreported in surveys.

## Conclusion

Our study revealed a positive decline in AFR across all socioeconomic and geographic groups. However, significant inequalities persist, with economic status and education level being the key drivers. The poorest quintile consistently experiences a much higher AFR compared to wealthier groups. This economic disparity contributes substantially to the overall burden of adolescent pregnancy. Whereas there was a higher decline in AFR over time among adolescent girls in terms of education, AFR increased among adolescent girls with both primary and secondary/higher levels of education between 2008 and 2019. This finding calls for the need to examine the kind of education provided for adolescent girls with at least primary education to better understand why they are resulting in an increase in AFR. While rural areas had a higher AFR than urban areas, this difference had a lesser impact on the overall national rate. Based on these findings, the government and policymakers in Sierra Leone should develop programs that improve economic opportunities and alleviate poverty, particularly for young women from disadvantaged backgrounds, which can significantly decrease their risk of early pregnancy. They should increase access to quality education, particularly for girls, that can equip young women with knowledge, skills, and agency to make informed choices about their reproductive health. Ensuring equitable access to comprehensive family planning services and education for all adolescents, regardless of location or socioeconomic background, is crucial. While regional variations are less significant, understanding and addressing specific factors contributing to higher AFR in certain provinces can further reduce national disparities. By implementing these recommendations and addressing the root causes of socioeconomic and educational inequalities, Sierra Leone can achieve a more equitable and significant reduction in adolescent pregnancy rates.

## Acknowledgments

We are grateful to MEASURE DHS and the World Health Organization for making the dataset and the HEAT software accessible.

## Author Contributions

**Conceptualization:** Augustus Osborne, Camilla Bangura, Bright Opoku Ahinkorah.

**Data curation:** Augustus Osborne, Camilla Bangura, Bright Opoku Ahinkorah.

**Formal analysis:** Augustus Osborne, Camilla Bangura, Bright Opoku Ahinkorah.

**Investigation:** Augustus Osborne, Camilla Bangura.

**Methodology:** Augustus Osborne, Camilla Bangura, Bright Opoku Ahinkorah.

**Supervision:** Bright Opoku Ahinkorah.

**Validation:** Bright Opoku Ahinkorah.

**Visualization:** Augustus Osborne, Camilla Bangura, Bright Opoku Ahinkorah.

**Writing – original draft:** Augustus Osborne, Camilla Bangura, Bright Opoku Ahinkorah.

**Writing – review & editing:** Augustus Osborne, Camilla Bangura, Bright Opoku Ahinkorah.

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
