## [Decision Letter · Decision Letter 0]

10 Oct 2024

PONE-D-24-34863Socio-economic and geographical inequalities in adolescent fertility rate in Sierra Leone,2008-2019PLOS ONE

Dear Dr. Osborne,

Thank you for submitting your manuscript to PLOS ONE. After careful consideration, we feel that it has merit but does not fully meet PLOS ONE’s publication criteria as it currently stands. Therefore, we invite you to submit a revised version of the manuscript that addresses the points raised during the review process.

While the two referees find the article suitable for publication with some minor revisions, note that the conclusion and, in particular, the policy section, present recommendations and analisys that do not stem from this study such as the discussion of microcredit, Please make sure to fully revise these sections excluding comments on factors that are not included in the empirical analysis. 

We look forward to receiving your revised manuscript.

Kind regards,

José Antonio Ortega, Ph.D.

Academic Editor

PLOS ONE

Journal Requirements:

2. Thank you for uploading your study's underlying data set. Unfortunately, the repository you have noted in your Data Availability statement does not qualify as an acceptable data repository according to PLOS's standards. At this time, please upload the minimal data set necessary to replicate your study's findings to a stable, public repository (such as figshare or Dryad) and provide us with the relevant URLs, DOIs, or accession numbers that may be used to access these data. For a list of recommended repositories and additional information on PLOS standards for data deposition, please see https://journals.plos.org/plosone/s/recommended-repositories.

Reviewers' comments:

Reviewer's Responses to Questions

**Comments to the Author**

1. Is the manuscript technically sound, and do the data support the conclusions?

Reviewer #1: Partly

Reviewer #2: Yes

2. Has the statistical analysis been performed appropriately and rigorously? 

Reviewer #1: Yes

Reviewer #2: Yes

3. Have the authors made all data underlying the findings in their manuscript fully available?

Reviewer #1: Yes

Reviewer #2: Yes

4. Is the manuscript presented in an intelligible fashion and written in standard English?

Reviewer #1: No

Reviewer #2: Yes

5. Review Comments to the Author

Reviewer #1: Explained responses

1. Is the manuscript technically sound, and do the data support the conclusions?

To some extent. Some conclusions and recommendations do not match the study findings.

Explanation is in specific comments

2. Has the statistical analysis been performed appropriately and rigorously?

Yes. Rigor has been displayed by using four metrics (D, R, PAF, and PAR) to establish disparity given the large sample sizes in the DHS datasets and the multiple inequality factors. However clarity is required in the methods section on the measurement and ranges of the metrics as explained in the specific comments

3. Have the authors made all data underlying the findings in their manuscript fully available?

Yes

4. Is the manuscript presented in an intelligible fashion and written in standard English?

-The authors need to innovatively present table 1 graphically so as to illustrate the trends for AFR across the three years, otherwise it does not quite come out in a tabular form

Below are some specific comments

1. Abstract, line 32

“Population attributable fraction decreased from -57.9 in 2008 to -23.9 in 2019” Did you mean to say decreased negativity as mentioned in previous text. Otherwise as it is now the value (PAF) increased between 2008 and 2019 regardless of being negative.

2. Introduction, line 48

The abbreviation AFR is used abruptly in the text without it full form.

3. Methods: Inequality measures, line 118

Given that this is a scholarly text, the word ‘lady’ should not be used in the text “The educational standing of the lady was categorized into four groups:” A more appropriate word can be used since women aged 15-19years were considered for the study

4. Methods: Statistical analysis, lines 125 – 134

- You mentioned in line 128 that the computation and analysis is in literature. However, before explaining the various interpretations of D, R, PAF, and PAR respectively, indicate the ranges of each to aid understanding, otherwise the text ends up being unclear without guiding the reader initially

5. Recommendations, lines 255 - 256

Just like in the abstract, You make recommendations on geographical location, “ensuring equitable access to family planning services and targeted interventions in rural areas are crucial steps towards achieving a more substantial and equal decline in national adolescent fertility rates.”. And yet in your findings inequalities were less likely to be influenced by location of residence hence a mismatch in your recommendation and your finding

Reviewer #2: The manuscripts addressed a very important topic in reproductive health among a vulnerable population. The exploration of inequalities in adolescent fertility rate will shed a light on a critical public health issue and has significant implications for policymakers, healthcare workers and researchers working on addressing adolescent reproductive health disparities.

6. PLOS authors have the option to publish the peer review history of their article (what does this mean?). If published, this will include your full peer review and any attached files.

Reviewer #1: No

Reviewer #2: No

---

## [Author Response · Author response to Decision Letter 0]

11 Oct 2024

The Editor

PLOS ONE

11th October 2024 

 Ref: PONE-D-24-34863

Title: Socio-economic and geographical inequalities in adolescent fertility rate in Sierra Leone,2008-2019

Response to Reviewers' comments 

Dear Sir/Madam, 

We want to express our sincere thanks for painstakingly reviewing our manuscript and providing valuable comments and suggestions. Please see our point-by-point response to the reviewers' comments and suggestions. Revisions are highlighted with track changes in the revised manuscript.

ACADEMIC EDITOR:

Thank you for submitting your manuscript to PLOS ONE. After careful consideration, we feel that it has merit but does not fully meet PLOS ONE’s publication criteria as it currently stands. Therefore, we invite you to submit a revised version of the manuscript that addresses the points raised during the review process.

While the two referees find the article suitable for publication with some minor revisions, note that the conclusion and, in particular, the policy section, present recommendations and analisys that do not stem from this study such as the discussion of microcredit, Please make sure to fully revise these sections excluding comments on factors that are not included in the empirical analysis. 

Response: Thank you. We have now revised that in the manuscript.

Journal Requirements:

Response: Thank you. We have done so.

2. Thank you for uploading your study's underlying data set. Unfortunately, the repository you have noted in your Data Availability statement does not qualify as an acceptable data repository according to PLOS's standards. At this time, please upload the minimal data set necessary to replicate your study's findings to a stable, public repository (such as figshare or Dryad) and provide us with the relevant URLs, DOIs, or accession numbers that may be used to access these data. For a list of recommended repositories and additional information on PLOS standards for data deposition, please see https://journals.plos.org/plosone/s/recommended-repositories.

Response: The dataset used can be accessed at https://dhsprogram.com/data/available-datasets.cfm

Reviewer #1: Explained responses

1. Is the manuscript technically sound, and do the data support the conclusions?

To some extent. Some conclusions and recommendations do not match the study findings.

Explanation is in specific comments

Response: Thank you. We have deleted that in the manuscript.

2. Has the statistical analysis been performed appropriately and rigorously?

Yes. Rigor has been displayed by using four metrics (D, R, PAF, and PAR) to establish disparity given the large sample sizes in the DHS datasets and the multiple inequality factors. However clarity is required in the methods section on the measurement and ranges of the metrics as explained in the specific comments

Response:Thank you. We have now revised to. The analysis was conducted using the online version of the HEAT software developed by the World Health Organisation (WHO) [30], which examines health disparities within and between nations across various health indicators and socioeconomic issues, including child, maternal, and reproductive health [34]. This study focused on four inequality metrics: difference (D), ratio (R), population-attributable fraction (PAF), and population-attributable risk (PAR). To clarify their interpretations, it is important to note the ranges of each metric: D can range from negative infinity to positive infinity, with a value of zero indicating no inequality and higher values reflecting greater inequality in adolescent fertility; R ranges from 1 to positive infinity, where a value of 1 signifies no inequality and values greater than 1 suggest increasing levels of adolescent fertility inequality; both PAF and PAR can take on values from negative infinity to positive infinity, with positive values indicating advantageous conditions and negative values reflecting unfavourable conditions, where greater magnitudes correspond to higher levels of inequality. The literature highlights the importance, computation, and analysis of these measures [34, 35]. In summary, higher values of D and R indicate greater inequality in adolescent fertility, while PAF and PAR values will be zero if no further progress can be made, meaning all subgroups have reached the same indicator level as the reference subgroup.

3. Have the authors made all data underlying the findings in their manuscript fully available?

Yes

Response: Thank you.

4. Is the manuscript presented in an intelligible fashion and written in standard English?

-The authors need to innovatively present table 1 graphically so as to illustrate the trends for AFR across the three years, otherwise it does not quite come out in a tabular form

Response: Thank you We have now provided a figure illustrating the trends in AFR across the three years.

Below are some specific comments

1. Abstract, line 32

“Population attributable fraction decreased from -57.9 in 2008 to -23.9 in 2019” Did you mean to say decreased negativity as mentioned in previous text. Otherwise as it is now the value (PAF) increased between 2008 and 2019 regardless of being negative.

Response: Thank you. Population attributable fraction increased from -57.9 in 2008 to -23.9 in 2019.

2. Introduction, line 48

The abbreviation AFR is used abruptly in the text without it full form.

Response: Thank you. We have now revised in the manuscript. Adolescent fertility rate (AFR)

3. Methods: Inequality measures, line 118

Given that this is a scholarly text, the word ‘lady’ should not be used in the text “The educational standing of the lady was categorized into four groups:” A more appropriate word can be used since women aged 15-19years were considered for the study

Response: We have now used the word women. Thank you.

4. Methods: Statistical analysis, lines 125 – 134

- You mentioned in line 128 that the computation and analysis is in literature. However, before explaining the various interpretations of D, R, PAF, and PAR respectively, indicate the ranges of each to aid understanding, otherwise the text ends up being unclear without guiding the reader initially

Response: We have now revised to. The analysis was conducted using the online version of the HEAT software developed by the World Health Organisation (WHO) [30], which examines health disparities within and between nations across various health indicators and socioeconomic issues, including child, maternal, and reproductive health [34]. This study focused on four inequality metrics: difference (D), ratio (R), population-attributable fraction (PAF), and population-attributable risk (PAR). To clarify their interpretations, it is important to note the ranges of each metric: D can range from negative infinity to positive infinity, with a value of zero indicating no inequality and higher values reflecting greater inequality in adolescent fertility; R ranges from 1 to positive infinity, where a value of 1 signifies no inequality and values greater than 1 suggest increasing levels of adolescent fertility inequality; both PAF and PAR can take on values from negative infinity to positive infinity, with positive values indicating advantageous conditions and negative values reflecting unfavourable conditions, where greater magnitudes correspond to higher levels of inequality. The literature highlights the importance, computation, and analysis of these measures [34, 35]. In summary, higher values of D and R indicate greater inequality in adolescent fertility, while PAF and PAR values will be zero if no further progress can be made, meaning all subgroups have reached the same indicator level as the reference subgroup.

5. Recommendations, lines 255 - 256

Just like in the abstract, You make recommendations on geographical location, “ensuring equitable access to family planning services and targeted interventions in rural areas are crucial steps towards achieving a more substantial and equal decline in national adolescent fertility rates.”. And yet in your findings inequalities were less likely to be influenced by location of residence hence a mismatch in your recommendation and your finding

Response: Thank you. We have deleted that in the manuscript.

Reviewer #2: The manuscripts addressed a very important topic in reproductive health among a vulnerable population. The exploration of inequalities in adolescent fertility rate will shed a light on a critical public health issue and has significant implications for policymakers, healthcare workers and researchers working on addressing adolescent reproductive health disparities.

Response: Thank you.

We hope that we have adequately addressed the reviewers' comments, and we look forward to receiving a favorable outcome on our paper. 

Yours Sincerely,

Augustus Osborne

Corresponding Author

---

## [Editor Report · Decision Letter 1]

17 Oct 2024

Socio-economic and geographical inequalities in adolescent fertility rate in Sierra Leone,2008-2019

PONE-D-24-34863R1

Dear Dr. Osborne,

We’re pleased to inform you that your manuscript has been judged scientifically suitable for publication and will be formally accepted for publication once it meets all outstanding technical requirements.

Kind regards,

José Antonio Ortega, Ph.D.

Academic Editor

PLOS ONE

Additional Editor Comments (optional):

The review has addressed the comment made by the reviewers and the editor and it has been felt unnecessary to send the revision back to the reviewers.
---

## [Editor Report · Acceptance letter]

2 Dec 2024

PONE-D-24-34863R1 

PLOS ONE

Dear Dr. Osborne, 

I'm pleased to inform you that your manuscript has been deemed suitable for publication in PLOS ONE. Congratulations! Your manuscript is now being handed over to our production team.

Kind regards, 

on behalf of

Dr. José Antonio Ortega 

Academic Editor

PLOS ONE